# AD-NEGF: An End-to-End Differentiable Quantum Transport Simulator for Sensitivity Analysis and Inverse Problems

## Abstract

Quantum transport theory describes transport phenomena from first principles, which is essential for domains such as semiconductor fabrication. As a representative, the Non-Equilibrium Green Function (NEGF) method achieves superiority in numerical accuracy. However, its tremendous computational cost makes it unbearable for high-throughput simulation tasks such as sensitivity analysis, inverse design, etc. In this work, we propose AD-NEGF, to the best of our knowledge the first Automatic Differentiation (AD) based quantum transport simulator. AD-NEGF calculates gradient information efficiently by utilizing automatic differentiation and implicit layer techniques, while guaranteeing the correctness of the forward simulation. Such gradient information enables accurate and efficient calculation of differential physical quantities and solving inverse problems that are intractable by traditional optimization methods.

## 1 Introduction

The strong and lasting demand for higher computing power and lower energy consumption urges the downscale of semiconductor devices. Over the last 40 years, the microelectronics industry has successfully made the transistor feature size scale from $10\mu$m to near 20nm, of which size the quantum mechanical effect starts to dominate (Anantram et al., 2008; Wang et al., 2008; Datta, 1997). Therefore, device simulators facing the future need to take a quantum theory oriented formulation, while NEGF, as a representative, is one of the most rigorous approaches among existing quantum transport methods (Jacoboni, 2010).

Although NEGF shows superiority in simulation accuracy, it is also extremely time and computation consuming. Recently, many works successfully integrate machine learning techniques to resolve the accuracy-efficiency dilemma of scientific simulations. A typical paradigm is to build up learning-based surrogate models (e.g., a neural network) (Li et al., 2020; Bürkle et al., 2021; Pimachev & Neogi, 2021). By learning from data generated with highly accurate simulations beforehand, the surrogate model is expected to maintain first-principle accuracy while performing much faster in usage. A fatal problem of such methods is that there is no guarantee for prediction accuracy, especially for input out of the distribution of the training dataset. Such drawback limits the application of machine learning based surrogates in quantum transport scenarios.

An alternative is to utilize automatic differentiation to make the computation process differentiable. In quantum transport simulations, practically useful information is often related to calculating derivatives. For instance, the thermoelectric property measured by the Seebeck coefficient; the sub-threshold swing of MOSFET that is related to the derivative of the drain current $I_D$ with respect to the applied gate voltage $V_g$, etc. Compared to traditional numerical differentiation, automatic differentiation can overcome the trade-off between the round-off error and the truncation error when choosing the step-size (Gautschi, 1997, Chap. 3), and also can be numerically more efficient when the input dimension is high. Moreover, in theoretical inverse problems, an end-to-end differentiable solver is also extremely useful and in fact, critical. The availability of gradients makes it possible to conduct efficient gradient-based optimization, which can outperform black-box optimization methods such as Bayesian optimization, genetic algorithm, etc., and can conduct optimization on a scale that black-box methods cannot. Recent advances have also shown the value to apply differentiable

programming in scientific computation scenarios, such as fluid dynamics (Holl et al., 2019), quantum chemistry (Kasim & Vinko, 2021), molecular dynamics (Schoenholz & Cubuk, 2020), photonic crystal optimization (Minkov et al., 2020), etc.

In this work, we propose AD-NEGF, to the best of our knowledge the first end-to-end differentiable quantum transport simulator. The entire numerical process of NEGF and TB modeling is implemented in PyTorch, including the computation of the self-energy term, the Green function, the electrostatic potential, the transport properties, as well as an optional Slater-Koster Tight-Binding (SKTB) module to generate the block tri-diagonal Tight-Binding (TB) Hamiltonian (Klymenko et al., 2021), which we will introduce in detail in Section 3. The backward pass to compute the gradients is improved by utilizing the implicit gradient techniques and the adjoint sensitivity method for Partial Differential Equations (PDE). To efficiently backpropagate through Poisson's equation in transport, we propose and implement the image charge gradient method, which can utilize the Fast Multi-pole Method (FMM) to reduce the backpropagation complexity of Poisson's equation from $O(N^3)$ to $O(N^{4/3})$. We demonstrate the capability of AD-NEGF to efficiently and accurately compute differential physical properties by comparing with numerical differentiation. Also, it is shown that by cooperating AD-NEGF with the gradient-based optimizer, it can perform high-dimensional optimization at a scale that is not affordable with conventional optimization approaches. Furthermore, in a more practical scenario of material doping optimization where we optimize the empirical SK parameters of injected atoms, our method shows significant advances in convergence speed and optimization solution, compared with traditional black-box optimization methods.

Our contributions can be summarized as follows:

- We propose and implement AD-NEGF, as far as we know the first end-to-end differentiable quantum transport simulator, including the NEGF method, the Poisson's equation module for self-consistent electrostatic potential computation, and the SKTB module to generate the tight-binding Hamiltonian from the coordinates and properties of the system atoms.

- The efficiency of the backward gradient computation is improved by applying the implicit gradient method, the adjoint method for PDEs, as well as our newly proposed gradient computation for the image charge method.

- We validate the advantages of AD-NEGF in calculating differential transport quantities, high-dimensional parameter fitting, and device optimization, where AD-NEGF outperforms numerical differentiation and black-box optimization methods.

## 2 RELATED WORKS

**NEGF.** Originating from Keldysh (1964); Kadanoff (2018), NEGF has been a well-received method in the quantum transport theory, which describes a system with a finite bias voltage and contact interactions under consideration. Recently, NEGF-based computation methods gain increasing popularity for the simplicity of the formulation, and the easy implementation in programming (Ferry & Goodnick, 1999; Taylor et al., 2001; Brandbyge et al., 2002; Fetter & Walecka, 2012), which makes NEGF one of the most widely applied methods in transport calculation. Several methods dedicated to improving its numerical stability and computational efficiency are proposed (Sancho et al., 1985; Krstić et al., 2002; Rungger & Sanvito, 2008), some of which are widely implemented in modern quantum transport simulation software, including but not limited to Papior et al. (2017); Smidstrup et al. (2019); Steiger et al. (2011). On the other hand, despite its advantages, the NEGF method suffers from heavy computational burdens.

**AI for Quantum Transport.** There have been prior works to apply machine learning techniques in quantum transport, mostly by training a neural network with data generated from first-principle simulations, so that the neural network can serve as an efficient surrogate model to predict transport properties, such as conductance (Bürkle et al., 2021; Pimachev & Neogi, 2021; Li et al., 2020), transport coefficients (Lopez-Bezanilla & von Lilienfeld, 2014), etc. Most existing methods use relatively simple deep learning models such as multi-layer perceptrons (Župančič et al.) and convolutional networks (Han et al., 2021; Souma & Ogawa, 2021; 2020), while in some cases more advanced and specially designed models are utilized (Bürkle et al., 2021). However, as mentioned in Section 1, a dataset generated with ab-initio simulation is required, which is expensive to obtain. Moreover,

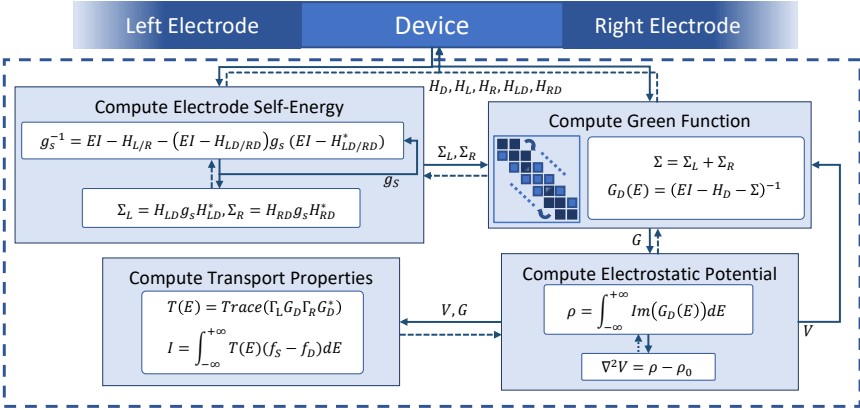

Figure 1: Workflow of AD-NEGF. Solid lines indicate the forward simulation flow, where loops denote self-consistent iterations. Dashed lines indicate the gradient backpropagation flow.

limited by the fundamental drawbacks of statistical learning, there is no guarantee for prediction accuracy, which will limit its application scenarios.

**Differentiable Programming.** Deep learning has been applied to more and more diverse scenarios, which requires the network structure to be more and more flexible. One emerging direction is to embed physical models or numerical computation processes into the model, in order to improve data efficiency, generalization capability, and interpretability. This is sometimes referred to as differentiable programming. It requires the automatic differentiation framework to support implicit numerical operations, such as fixed-point iterations (Bai et al., 2019), optimization (Amos & Kolter, 2017), initial value problems (Chen et al., 2018), etc. Differentiable programming has been widely applied to physical simulations (Hu et al., 2019; Innes et al., 2019), such as rigid body dynamics (de Avila Belbute-Peres et al., 2018; Freeman et al., 2021), computational fluid dynamics (Kochkov et al., 2021; Holl et al., 2019; Schenck & Fox, 2018), ray tracing (Li et al., 2018), etc. More specifically in ab-initio simulations, there have been works for density functional theory (Li et al., 2021; Kasim & Vinko, 2021), Hartree–Fock (Tamayo-Mendoza et al., 2018), coupled cluster methods (Pavošević & Hammes-Schiffer, 2020), and molecular dynamics (Schoenholz & Cubuk, 2020). However, we have not found any previous works to apply differentiable programming techniques in the quantum transport domain.

## 3 PRELIMINARIES FOR THE NON-EQUILIBRIUM GREEN FUNCTION METHOD

In this section, we give a brief introduction to the NEGF method, while more details can be referred to in Appendix A. Consider a transport system containing a device region and two semi-infinite contacts that attach to the left and right sides of the device, as shown in Figure 1. The contacts can also be referred to as leads or electrodes interchangeably. According to the theory of quantum mechanics, the whole system, including the device and the contacts, can be fully described by its Hamiltonian $H$. In this paper, we consider the Tight-Binding (TB) model (Slater & Koster (1954)), which makes $H$ block tri-diagonal. We assume a set of basis has been selected so that the full NEGF process can be expressed in the matrix form. The stationary Schrödinger equation of this open system is $H\Psi = E\Psi$, where $\Psi$ stands for the wave function of electrons, and $E$ is a scalar value corresponding to the system energy. The characteristics of the system are contained in its Green function

$$G = [EI - H]^{-1}, \tag{1}$$

where $I$ is the identity matrix. However, the Hamiltonian $H$ is infinitely large and hence intractable. This is resolved by computing the Green function only for the device part, and modeling the effect of two semi-infinite contacts in a term $\Sigma$ referred to as self-energy. The device Green function $G_D$ will then be used to describe the non-equilibrium charge transport process by solving Poisson's equation

in a self-consistent iteration. The output self-consistent potential field $V$ and $G_D$ can be used to compute transport properties, such as transmission, current, etc.

**Device Green Function.** The device Green function is expressed as a function of the device Hamiltonian $H_D$ and the self-energy $\Sigma$:

$$G_D = [EI - H_D - \Sigma]^{-1}. \tag{2}$$

Directly computing the matrix inversion is with complexity $O(N^3)$, which is unbearable as the matrix size is proportional to the vast amount of atoms. By utilizing the block tri-diagonal form of the Hamiltonian matrix, an efficient recursive algorithm (Anantram et al., 2008) can be implemented, which scales linearly with the system size.

**Electrode Self-Energy.** Self-energy of electrodes is computed from the surface green function $g_s$ of the electrode layer coupled with devices. Under the half-infinite hypothesis, $g_s$ is approximated identically, expressed as a self-consistent equation:

$$g_s^{-1} = [A_l - A_{l,l-1} g_s A_{l-1,l}^{\dagger}], \tag{3}$$

where $A_{l,l-1}$ is blocks of $EI - H$ of coupling between $l$ and $l-1$ layer. To speed up, we implement the Lopez-Sancho algorithm (Sancho et al., 1985), as illustrated in Algorithm 1, which converges exponentially faster than the conventional self-consistent iteration. Details of the algorithm are illustrated in Appendix. We also implement a modern method based on the generalized eigenvalue problem (Wang et al., 2008) as an alternative.

**Electrostatic Potential.** In NEGF, charge transfer due to the applied bias voltage is modeled as an external potential, which is attained self-consistently by solving Poisson's equation for the electrostatic field:

$$\begin{cases} \nabla \cdot \epsilon(r) \nabla [\Delta V(r)] &= -[\rho(r; \Delta V) - \rho_0(r)], \\ \Delta V(r)|_{\{z_L, z_R\}} &= \{V_L, V_R\}. \end{cases} \tag{4}$$

where $V_L$ and $V_R$ represent the voltage boundary conditions at electrodes $z_L$ and $z_R$, $\Delta V = V - V_0$ is the difference between real potential energy with equilibrium one. This equation is solved self-consistently with updated $H_{neq}$. Poisson's equation can be solved using numerical PDE solvers with spherical charges. Meanwhile, a computationally more efficient image charge method using the Fast Multipole Method (FMM) is preferred (Svizhenko & Anantram, 2005; Zahn, 1976). After the procedure converges to a stable solution, transport properties can be computed accordingly. Once the convergence is achieved, the Green function computed with $H_{neq}$ will be used to compute various transport properties. We refer to Appendix for details of equation solving, and expression of transport properties.

## 4 METHOD OF DIFFERENTIATING THE NEGF PROCESS

The differentiable NEGF model is implemented with PyTorch (Paszke et al., 2019). We extend the autograd function with implicit gradient techniques for backpropagation through self-consistent iterations, with the adjoint sensitivity method for calculating gradients through Poisson's equation (Pontryagin, 1987). Moreover, the efficient gradient formula for the image charge method (Svizhenko & Anantram, 2005), accelerated by the Fast Multipole Method (FMM) method, is proposed to speed up the gradient calculation of Poisson's equation. The derived formula can be regarded as a summation of point charges produced by the gradients and thus can also be computed with FMM. Details of the customized backward propagation modules are explained as follows.

### 4.1 IMPLICIT GRADIENT

The implicit gradient method is implemented when the direct automatic differentiation through function $y = f(x)$ is unavailable or expensive to compute. Instances often arise when one wants to calculate gradients through numerical solvers of equilibrium problems or complicated iterative algorithms. Based on the implicit function theorem (Krantz & Parks, 2002), if there exists such

constrained function $h(y, x) = 0$ where $y$ is taken as the converged output of function $f$, the gradient $\frac{dy}{dx}$ can be given as $\frac{dy}{dx} = -\left[\frac{\partial h(y,x)}{\partial y}\right]^{-1}\frac{\partial h(y,x)}{\partial x}$.

We use the implicit gradient techniques to derive the gradient of the surface Green function (Sancho et al., 1985), where according to the ideal definition of the two semi-infinite leads, the converged surface Green function $g_s(\theta)$ must satisfy the self-consistent Equation (3). Hence $h(g_s, \theta) = [A_{ll} - A_{ll-1}g_s A^{\dagger}_{l-1l}] - gs^{-1} = 0$, where $A_{ll}$ stands for $[ES_{ll} - H_{ll}]$, and $\theta$ denotes the input variables to compute $g_s$. Thus we could write down the gradient of $g_s$ with respect to $\theta$ explicitly by $\frac{dg_s}{d\theta} = -\left[\frac{\partial h(g_s,\theta)}{\partial g_s}\right]^{-1}\frac{\partial h(g_s,\theta)}{\partial \theta}$.

Another scenario that the implicit gradient method can be applied to is to compute gradients through the self-consistent Poisson's equation loop, where the system electrostatic potential is updated until consistent with the bias voltage of contacts and other boundary conditions.

## 4.2 ADJOINT METHOD FOR PDE

In order to perform backpropagation through the solver for Poisson's equation, adjoint sensitivity method (Plessix, 2006; Pontryagin, 1987) for PDE-constrained optimization problems is adopted, which is widely applied in constrained optimization of inverse problems. Here, the forward process of the numerical PDE solver is unaltered, which is often denoted as the state equation that links the controlled parameter and the state of the constrained system. Meanwhile, an adjoint state equation that connects the perturbation of variables and states is solved by using the same numeral solver. Then gradients can be evaluated with the adjoint state, and join in the gradient chain of backward propagation. Since the adjoint state equation is independent of the number of controlled variables, the total complexity is only proportional to the forward process, which makes it suitable for control problems with scalar output and high-dimensional input. Recently, the adjoint sensitivity method has also been applied in designing neural network structures with physics intuitions, including the Neural ODE (Chen et al., 2018) and Deep Equilibrium Models (Bai et al., 2019), which can be considered as examples of cooperations of automatic differentiation and adjoint methods.

## 4.3 GRADIENT OF FMM IMAGE CHARGE METHOD

An alternative approach to solve Poisson's equation raised in Equation (4), is to apply the point charge approximation, where the charge density is considered as the linear combination of a series of point charges as $\Delta q(r) = \sum_i \Delta q_i \delta(r - r_i)$. Then by employing the linearity of Poisson's equation, the original form can be further decomposed into Laplace's equation with Dirichlet boundary conditions and Poisson's equation with zero Dirichlet boundary conditions:

$$\begin{cases} -\nabla^2(\Delta V_1(r)) = 0, \\ \Delta V_1(r)|_{\{z_L, z_R\}} = \{V_L, V_R\}. \end{cases} \qquad \begin{cases} -\nabla^2(\Delta V_2(r)) = \frac{1}{\epsilon}\Delta\rho(r), \\ \Delta V_2(r)|_{\Sigma} = 0. \end{cases} \tag{5}$$

The first Laplace's equation can be easily solved by a linear drop potential. The second equation can be solved by assuming the charge density as a combination of point charges of each atom site. The closed-form solution can be obtained using the image charge method (Svizhenko & Anantram, 2005; Harb, 2019), and the second potential can be written as:

$$V_2(r_i) = \sum_{j \in N, j \neq i} \frac{q_j}{4\pi\epsilon} \frac{1}{\sqrt{t_{ij}^2 + (z_i - z_j)^2}}$$

$$+ \sum_{j \in N} \frac{q_j}{4\pi\epsilon} \sum_{n=1}^{\infty} \left[ \frac{1}{\sqrt{t_{ij}^2 + \Delta_1^2}} - \frac{1}{\sqrt{t_{ij}^2 + \Delta_2^2}} + \frac{1}{\sqrt{t_{ij}^2 + \Delta_3^2}} - \frac{1}{\sqrt{t_{ij}^2 + \Delta_4^2}} \right], \tag{6}$$

where $t_{ij}^2 = (x_i - x_j)^2 + (y_i - y_j)^2$, and $\Delta^2$ stands for the distance in the transport direction between central charges and charges from two electrodes. Therefore, the first term here describes the interactions inside the device, while all the remaining terms simulate the effect of its coupling to charges outside. The summation of the second term is computed until achieving certain accuracy, which is empirically hundreds of site numbers. Hence a direct summation is also too expensive to

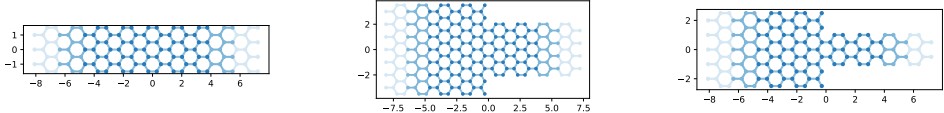

(a) Structure of an AGNR with width 7 and length 5.

(b) Structure of a 7-4 graphene nano-junction.

(c) Structure of a 5-2 graphene nano-junction.

Figure 2: Device structures used in the experiments.

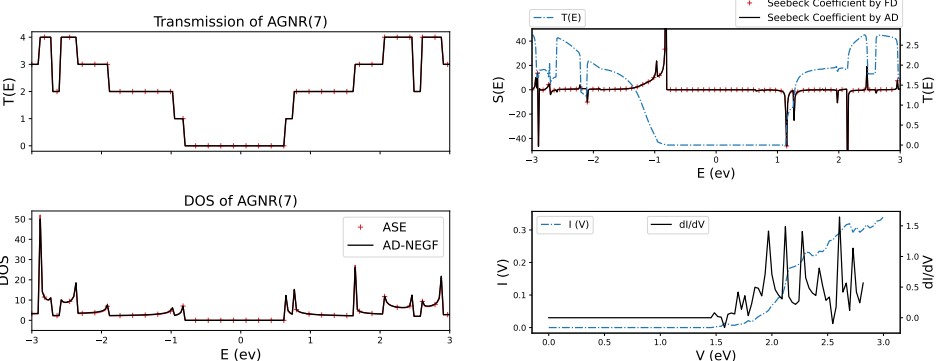

(a) Transmission and DOS calculated by AD-NEGF and confirmed with ASE.

(b) Seebeck coefficient and differential conductance calculated by AD-NEGF.

Figure 3: Transmission Quantity Computation with AD-NEGF.

compute. In this case, the Fast Multipole Method (Engheta et al., 1992) is employed to reduce the computational complexity from $O(N^3)$ to $O(N^{4/3})$.

To perform backward propagation through the fast multipole layer, the gradient of the output potential to the charges is required. By taking the derivative of a target objective $L : C^d \to R$, the derivative of $L$ with respect to charge $q_j$ can be expanded as the image summation form of accumulated gradients from the last layer, which is:

$$\frac{\partial L(V)}{\partial q_j} = \sum_i \frac{\partial L}{\partial V_i} \frac{\partial V_i}{\partial q_j} \tag{7}$$

$$= \sum_{i \in N, i \neq j} \frac{\partial L / \partial V_i}{4\pi\epsilon} \frac{1}{\sqrt{t_{ij}^2 + (z_j - z_i)^2}}$$

$$+ \sum_{i \in N} \frac{\partial L / \partial V_i}{4\pi\epsilon} \sum_{n=1}^{\infty} \left[ \frac{1}{\sqrt{t_{ij}^2 + \Delta_1^2}} - \frac{1}{\sqrt{t_{ij}^2 + \Delta_2^2}} + \frac{1}{\sqrt{t_{ij}^2 + \Delta_3^2}} - \frac{1}{\sqrt{t_{ij}^2 + \Delta_4^2}} \right]. \tag{8}$$

Similarly, computing gradients of this form can be accelerated by the Fast Multipole Method, which is also with complexity $O(N^{4/3})$ and much faster than solving adjoint Poisson's equation.

## 5 APPLICATIONS

In this section, the advantages of AD-NEGF for sensitivity analysis and inverse problems are demonstrated with three applications. For all experiments, we take graphene as the transport system, including the Armchair Graphene NanoRibbon (AGNR) and the graphene nano-junction, the basic device structures of which are displayed in Figure 2. More details of the experimental setup can be found in Appendix B.

## 5.1 DIFFERENTIAL TRANSMISSION QUANTITY COMPUTATION

A direct and major application to perform differentiation on physical models is to evaluate differential physical quantities. In most cases, the analytical form is difficult to obtain. For numerical differentiation, the trade-off will be encountered between the round-off error and the truncation error when choosing the step-size (Gautschi, 1997, Chap. 3), and the computation will be unacceptable when the input dimension is high. On the contrary, automatic differentiation can achieve machine precision while maintaining $O(1)$ complexity when the output dimension is low and the input dimension is high (Baydin et al., 2018).

In this experiment, we first validate the correctness of the forward computation of AD-NEGF. As shown in Figure 3(a), the transmission coefficient and the density of states (DOS) of an AGNR system with width 7 are computed by AD-NEGF, which perfectly match the results of ASE (Larsen et al., 2017), an atomistic simulation package including electron transport modules. Based on it, we compute two differential transmission quantities, the Seebeck coefficient and the differential conductance, which are shown in Figure 3(b). The Seebeck coefficient is a measure of the magnitude of an induced thermoelectric voltage in response to a temperature difference across an atomic structure, mathematically expressed as the derivative of transmission $T(E)$ concerning the chemical potential $E$ (Reddy et al., 2007):

$$S_{junction} = -\frac{\pi^2 k_B^2 T}{3e} \frac{\partial ln(T(E))}{\partial E},  \tag{9}$$

where $T$ stands for the temperature and $k_B$ is the Boltzmann constant. The differential conductance is the gradient of electronic current to voltage: $I_D = \frac{dI}{dV}$.

The singularity of the transmission function leads to peaks in the Seebeck coefficient curve, which is highly sensitive thus challenging for derivative calculation, as illustrated in Figure 4. To amplify the phenomenon for clearer demonstration, the output transmission coefficient $T(E)$ of the forward computation is transformed into half-precision floating-point format for both automatic and numerical differentiation, before it is used to compute the Seebeck coefficient. It can be seen that, with AD-NEGF, we can still generate high-quality results. However, for numerical differentiation, the trade-off between the truncation error and the round-off error is observed by selecting different step-sizes from 1e-2 to 1e-5. With a large step-size, peaks may be skipped or mistakenly generated due to truncation error. With a small step-size, lacking machine precision causes noises on the curve. Specifically for step-size 1e-5, the calculated curve becomes totally meaningless. Moreover, even though this is not a high-dimensional input situation, evaluating the Seebeck coefficient with AD-NEGF can still be faster than numerical differentiation, since in AD-NEGF the backward pass is improved. According to our experiments, for a smaller system with 70 carbon atoms, computing the Seebeck coefficient for 400 energy samples costs 71.1 seconds with AD-NEGF and 98.3 seconds with numerical differentiation. For a larger system with 240 carbon atoms, computing the Seebeck coefficient for 400 energy samples costs 363.1 seconds with AD-NEGF and 512.6 seconds with numerical differentiation.

To summarize, by conducting the above experiments, the correctness and effectiveness of AD-NEGF are validated. With AD-NEGF, differential transport quantities can be calculated simply by calling one backward step. Moreover, the process of computing derivatives is itself differentiable, permitting the computation of higher-order derivatives, which remains for further discovery.

## 5.2 TRANSMISSION FITTING

Inverse problems, which require inferring input parameters reversely from the output objectives, are in general difficult in first-principle simulations. Black-box optimization methods require sampling a large number of input combinations, the cost of which grows exponentially with the number of parameters. Based on the efficient and accurate gradient computation capability of AD-NEGF, performing gradient-based optimization holds the potential to outperform black-box optimization methods for high dimensional inverse problems.

We conduct a $10^4$ dimensional optimization experiment to fit the transmission curve of one graphene nano-junction to another. The target system is a 7-4 nano-junction, with 7 graphene rings on the left and 4 on the right. The fitting system is a 5-2 nano-junction, and the fitting variables are the elements of its Hamiltonian, including the device, leads, and the corresponding couplings. The dimension of the optimizing variables is at the level of $10^4$. The transmission curve, as shown in Figure 5,

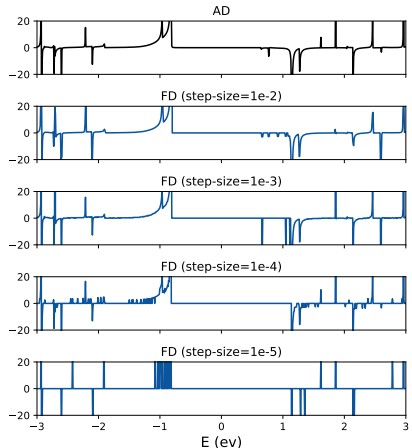 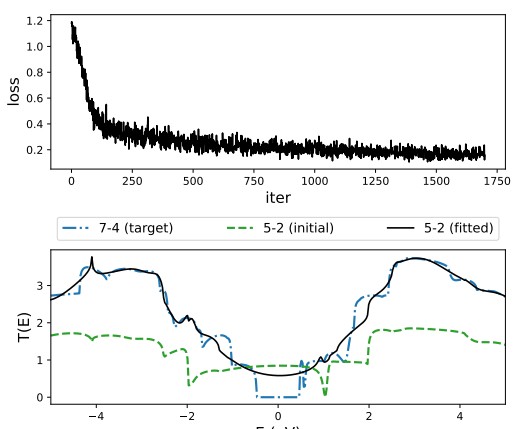

Figure 4: Comparison of Automatic Differentiation and Numerical Differentiation with different step-sizes.

Figure 5: The fitting loss and the fitted transmission curve of a 5-2 graphene nano-junction.

consists of 2000 energy points sampled from (-5eV, 5eV). Since directly computing the gradients of all 2000 points is inefficient for iterations, we apply the stochastic gradient descent algorithm to conduct mini-batch optimization, which has shown supremacy of efficiency and performance in high dimensional optimization problems. More specifically speaking, the fitting parameters are optimized with the Adam optimizer (Kingma & Ba, 2014) built in PyTorch, making the procedure highly similar to training a neural network.

The results are displayed in Figure 5, where the loss is reduced to a considerably low level, which means the converged parameters of the 5-2 nano-junction fit nicely to the larger 7-4 nano-junction. The fitted curve is akin to a smoothed version of the curve of the 7-4 junction, which agrees with the intuition since a graphene junction of 5-2 is of less freedom than that of a 7-4 nano-junction. On the other hand, we have also tried traditional black-box methods, such as Bayesian optimization, the genetic algorithm, and gradient-based optimization with numerical differentiation, but none of them can even work for this problem because of the curse of dimensionality.

### 5.3 ON-SITE DOPING OPTIMIZATION

Modern material engineering is capable of manipulating at the atomic level. More specifically speaking, by performing processes such as deformation, doping, etc., microscopic physical quantities such as atomic spatial coordinates, bond lengths and doping positions can be changed, which further modify the macroscopic material properties. The doping process is one of the most common techniques in material development, which can dramatically change the properties of the original material, by injecting foreign atoms into specific positions. In this experiment, we further explore the possibility to solve practical inverse problems with AD-NEGF by performing an end-to-end doping optimization cooperated with established material models.

In this experiment, we try to reduce the average transmission of an AGNR system in a specified energy range of (-1eV, 1eV), by injecting other atoms into the center of the AGNR system along the transmission direction. A reduction of transmission coefficient near zero Energy point would indicate an increase of the truncation voltage, which changes the semi-conductive properties of the device (Wu et al., 2013). Doping can be modeled as an effective change in the site and the hopping terms in the tight-binding Hamiltonian, i.e., the diagonal and off-diagonal elements of the Hamiltonian matrix. This on-site approximation allows us to treat doping optimization as tuning local terms in the Hamiltonian influenced by the injected atoms. However, although the process above is applicable, the tuning terms in the TB Hamiltonian need to be distinguished carefully from those invariant ones. It will be more convenient to cooperate with an SKTB model, which constructs the TB Hamiltonian based on strict rules of local dependence of atom identities and their semi-empirical SK parameters. Besides convenience, it has more concrete physical interpretation than directly optimizing elements

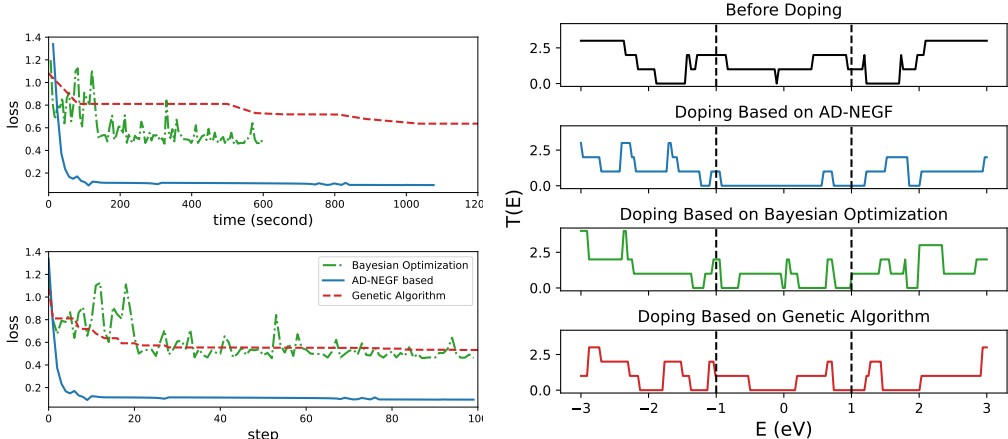

(a) Loss against running time and iteration steps respec-   (b) Original and optimized transmission curves.
tively.

Figure 6: Comparison between AD-NEGF and conventional black-box optimization methods in the doping optimization task.

of the Hamiltonian, since it provides guidelines for practitioners to find the possible atom satisfying the SKTB parameters from the optimization result. In this way, doping optimization is modeled as an optimization of the SKTB parameters of the doped atoms, which include the orbital energy and parameters for two center integrals. The total number of optimization variables is 13.

For comparison, we also apply black-box optimization methods including the genetic algorithm and Bayesian optimization. The results are displayed in Figure 6. In the loss diagram, the gradient-based method converges significantly faster and better than the other approaches, especially at the beginning of the training. The loss curves of the genetic optimization and Bayesian optimization are also dropping, but much slower and less effective, in terms of either the running time or the iteration step. Moreover, the performances of the genetic / Bayesian optimization are sensitive to preset hyper-parameters. Corresponding to the loss curves, the results of optimized transmission curves demonstrate the advantages of AD-NEGF in a more straightforward way, where the gradient-based optimization gives a much cleaner band with low transmission in the target interval (-1eV, 1eV) compared to other methods. These results validate the effectiveness of the AD-NEGF method in conducting practical atomic-level inverse design to optimize transport properties by cooperating with material models.

## 6  CONCLUSION

In this paper, we have proposed AD-NEGF, the first end-to-end differentiable quantum transport simulator to the best of our knowledge. It guarantees the correctness of the forward simulation without the need for data or training, while providing gradient information based on differentiable programming. Compared with numerical differentiation, gradients can be computed more efficiently and accurately. Moreover, it accelerates parameter fitting and parameter optimization with gradient-based optimization. The results are validated in applications such as differential physical quantity computation, transmission curve fitting, and device doping optimization.

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

# A    ADDITIONAL DETAILS ON THE NEGF METHOD

## A.1    COMPUTATION OF THE SELF-CONSISTENT SURFACE GREEN FUNCTION

Since the system is made up of a device and two semi-infinite contacts on the side, Equation (1) can be expanded in the following form:

$$\begin{bmatrix} A_L & A_{LD} & 0 \\ A_{DL} & A_D & A_{DR} \\ 0 & A_{RD} & A_R \end{bmatrix} \begin{bmatrix} G_L & G_{LD} & G_{LR} \\ G_{DL} & G_D & G_{DR} \\ G_{RL} & G_{RD} & G_R \end{bmatrix} = I, \tag{10}$$

where $A = [EI - H]$, and the subscripts are used to distinguish the matrix elements corresponding to the left lead (L), the device (D), the right lead (R), and their interactions. Thanks to its block tri-diagonal form, the device Green function $G_D$ satisfies

$$[A_D - A_{DL}A_L^{-1}A_{LD} - A_{DR}A_R^{-1}A_{RD}]G_D = I. \tag{11}$$

Since $A_D = [EI - H_D]$, compared with Equation (2), we have

$$\Sigma^L = A_{DL}A_L^{-1}A_{LD}, \tag{12}$$

$$\Sigma^R = A_{DR}A_R^{-1}A_{RD}, \tag{13}$$

$$\Sigma = \Sigma^L + \Sigma^R. \tag{14}$$

Here we assume only the neighbouring layers have interactions with each other, and denote the left lead layer connected to the device by $l$. Then the left self-energy can be simplified as $\Sigma^L = A_{Dl}A_l^{-1}A_{lD}$. The coupling matrix $A_{lD}$ is given as input of NEGF. What remains unclear is $A_l^{-1}$, the bottom-right block of $A_L^{-1}$. This is known as the surface green function, denoted as $g_s$. By utilizing the ideal lead assumption that removing one layer of the lead will not change $g_s$ obtain a self-consistent form as 3, where Lopez-Sancho algorithm (Sancho et al., 1985) can be applied to accelerate the convergence, here we display the detailed algorithm below:

---

**Algorithm 1** Lopez-Sancho algorithm for surface Green function

---

set $\epsilon_0^s = h_{0,0}, \epsilon_0 = h_{0,0}, \alpha_0 = h_{0,1} - ES_{0,1}, \beta_0 = h_{1,0} - ES_{1,0}$
**repeat**
  $\epsilon_i^s = \epsilon_{i-1}^s + \alpha_{i-1}(ES - \epsilon_{i-1})^{-1}\beta_{i-1},$
  $\epsilon_i = \epsilon_{i-1} + \beta_{i-1}(ES - \epsilon_{i-1})^{-1}\alpha_{i-1} + \alpha_{i-1}(ES - \epsilon_{i-1})^{-1}\beta_{i-1}$
  $\alpha_i = \alpha_{i-1}(ES - \epsilon_{i-1})^{-1}\alpha_{i-1}$
  $\beta_i = \beta_{i-1}(ES - \epsilon_{i-1})^{-1}\beta_{i-1}$
**until** converge
$g_{0,0} = (ES - \epsilon_m^s)^{-1}$

---

## A.2    COMPUTATION OF THE SELF-CONSISTENT ELECTROSTATIC POTENTIAL

Denote the charge densities in the equilibrium and non-equilibrium states as $\rho_0$ and $\rho$, and the potential fields from the original neutral and redistributed charges as $V_0$ and $V$. The equilibrium and non-equilibrium Hamiltonian can be expressed as $H_0 = T + V_0$, $H_{neq} = T + V$, where $T$ is the kinetic energy. Poisson's equation relates potentials to the corresponding charge densities:

$$\begin{cases} \nabla \cdot \epsilon(r)\nabla V(r) = -\rho(r), \\ \nabla \cdot \epsilon(r)\nabla V_0(r) = -\rho_0(r). \end{cases} \tag{15}$$

Therefore we have $\nabla \cdot \epsilon(r)\nabla[\Delta V(r)] = -[\rho(r) - \rho_0(r)]$, where $\Delta V = V - V_0$ is used to correct the Hamiltonian by $H_{neq} = H_0 + \Delta V$. The updated $H_{neq}$ will again be used to update $\Delta V$. Hence a self-consistent iteration is constructed:

$$\begin{cases} \nabla \cdot \epsilon(r)\nabla[\Delta V(r)] &= -[\rho(r; \Delta V) - \rho_0(r)], \\ \Delta V(r)|_{\{z_L, z_R\}} &= \{V_L, V_R\}. \end{cases} \tag{16}$$

Charge densities are necessary input for the above equation. Denote potentials in left and right electrodes as $u_l$ and $u_r$ (assume $u_l < u_r$), then the charge density $\rho(r) = -\frac{i}{2\pi}\int_{-\infty}^{+\infty} dE G(E)$, which can be decomposed into equilibrium and non-equilibrium terms:

$$\rho(r) = \rho_{eq}(r) + \rho_{neq}(r) \tag{17}$$

$$= \frac{1}{\pi}Im\left[\int_{-\infty}^{u_l} dE G_D(E)\right] + \frac{1}{2\pi}\int_{u_l}^{u_r} dE G_D(E). \tag{18}$$

The first integration up to infinity can be computed efficiently using contour integration with the residue theorem. It is achieved by expanding the Fermi-Dirac function (Ozaki, 2007; Areshkin & Nikolić, 2010). On the other hand, the non-equilibrium charge density $\rho_{neq}$ is computed directly by numerical integration. The density of neutral charges $\rho_0$ can be computed by setting $u_l = u_r = 0$.

### A.3 EXPRESSIONS OF TRANSPORT PROPERTIES

With the NEGF theory, electronic transport properties can be derived, such as the transmission probability ($T(E)$), the density of states ($DOS$), the electronic current ($I$), the equilibrium and non-equilibrium electronic densities ($\rho_{eq}$ and $\rho_{neq}$), etc. Here we list some of the expressions:

$$T(E) = Trace[\Gamma_L(E)G_D(E)\Gamma_R(E)G_D^\dagger(E)], \tag{19}$$

$$DOS(E) = -\frac{1}{\pi}Trace[Im(G_D(E))], \tag{20}$$

$$I = \frac{2e}{\hbar}\int_{-\infty}^{+\infty}\frac{dE}{2\pi}T(E)[f(E - u_l) - f(E - u_r)], \tag{21}$$

$$\rho(r) = \frac{1}{\pi}Im\left[\int_{-\infty}^{u_l} dE G_D(E)\right] + \frac{1}{2\pi}\int_{u_l}^{u_r} dE G_D(E). \tag{22}$$

For Equation (21), the integral range of the current is decided by the subtraction of the Fermi-Dirac function, which is a little wider than $(u_l, u_r)$.

## B ADDITIONAL DETAILS ON EXPERIMENTAL SETUP

The experiments are run on an Intel(R) Xeon(R) CPU E5-2650 v4 @ 2.20GHz CPU, and an NVIDIA Tesla P40 GPU. We implemented our method in PyTorch 1.9.1. We validated the correctness of our simulation results by comparing with ASE of version 3.22.0.

In the experiments, we set the learning rate of the Adam optimizer as 0.001, and the batch size as 64. Bayesian optimization is implemented based on Nogueira (2014–), and the genetic algorithm is implemented based on Solgi (2020–). The bounds of the optimization variables for the black-box optimizers are $(\theta_0 - 0.3, \theta_0 + 0.3)$, where $\theta_0$ is the initial value, namely the original 5-2 nano-junction TB Hamiltonian for the transmission curve fitting experiment, and undoped SKTB parameters for the device doping optimization experiment.

The hyper-parameters of the genetic algorithm are:

```
{
    "max_num_iteration": None,
```

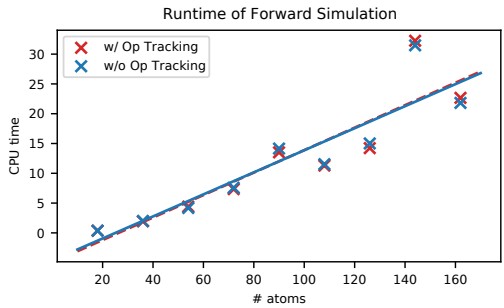

(a) Forward simulation runtime cost against the system size.

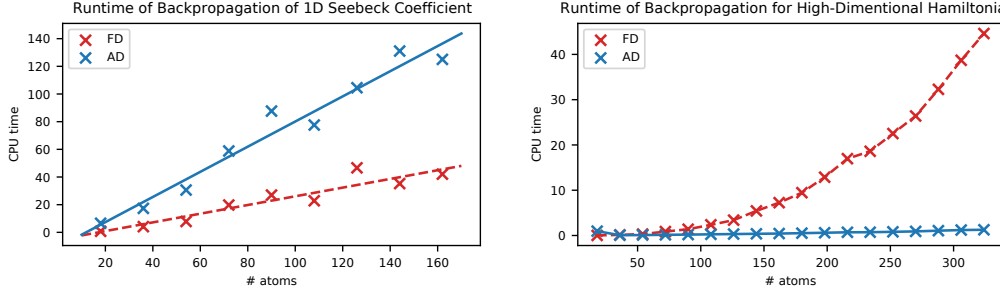

(b) Backward 1D derivative computation runtime cost against the system size.

(c) Backward multi-dimensional derivative computation runtume cost against the system size.

Figure 7: Runtime cost of forward and backward computation.

```
    "population_size": 20,
    "mutation_probability": 0.1,
    "elit_ratio": 0.01,
    "crossover_probability": 0.5,
    "parents_portion": 0.3,
    "crossover_type": 'uniform',
    "max_iteration_without_improv": None
}
```

The hyper-parameters of the Bayesian Optimization algorithm are:

```
{
    "random_state": 3,
    "verbose": 2,
    "kind": "ucb",
    "kappa": 2.5
    "xi": 0.0
}
```

We have uploaded our source code in the supplementary materials for cross-checking. The code will also be released and maintained as an open-source repository in the future.

## C  SCALABILITY AND RUNTIME ANALYSIS OF AD AND FD

In this section, we present the test results and analysis of the scalability and runtime for both the forward simulation and the backward gradient calculation of AD-NEGF, compared against traditional numerical differentiation. The test system we use is AGNR. By controlling the length of AGNR, the number of atoms contained in the system changes accordingly, so that we can test the performances

with respect to the system scale. The time cost is measured by the CPU time with functions provided by the python standard library.

In the forward simulation test, we compute the transmission function $T(E)$ the same as in Section 5. Since a differentiable solver requires tracking each conducted operation in the automatic differentiation tool, which brings extra computational cost compared to conventional simulation programs, and turning off such operation tracking functionality will make the differentiable solver behave just like a traditional solver in forward simulation, we compare the CPU time cost for AGNR systems in different sizes, with both the operation tracking functionality turned on, and with the functionality turned off. The results are displayed in Figure 7(a), from which we can conclude that, the CPU time cost scales linearly with the system size, and the additional computational cost brought by operation tracking is almost negligible.

In the backpropagation test, we compare the runtime cost of calculating derivatives for both 1-dimensional variables (the Seebeck coefficient) and multi-dimensional variables (device Hamiltonians). The results for both cases are shown in Figure 7(b) and 7(c) respectively. In the Seebeck coefficient calculation experiment, the derivative is computed as $\partial T(E)/\partial E$, where we sample 100 energy points in each run. We can see the computational cost of both FD and AD scales linearly with the system size. Since the backpropagation computation is more complex than the forward computation, the slope of AD is a bit larger than FD. However, in such cases, efficiency will not be the bottleneck anyway, while as demonstrated in Section 5, AD can achieve higher precision than FD. For the high-dimensional variable gradient computation, as shown in Figure 7(c), we can see AD significantly outperforms FD in computing gradients for the device Hamiltonians of dimension $N_A^2$, where $N_A$ is the number of atom orbitals. This is because, FD can only compute the derivative for one dimension of the input variable each time, making its complexity grow in $O(N_A^2)$. On the other hand, the complexity to compute gradients by backward-mode AD depends only on the complexity of the forward simulation, regardless of the increasing input variable dimensions.

In conclusion, by utilizing differentiable programming techniques, AD-NEGF, as an end-to-end differentiable quantum transport simulator, computes gradients for high-dimensional variables significantly faster than FD, while merely increasing the forward simulation cost.

