# OpenReview forum: "AD-NEGF: An End-to-End Differentiable Quantum Transport Simulator for Sensitivity Analysis and Inverse Problems"
_ICLR.cc/2023/Conference — Submitted to ICLR 2023_

### Official Review · Reviewer_4fBj · 2022-10-25

**Confidence:** 2
**Correctness:** 3
**Technical Novelty And Significance:** 3
**Empirical Novelty And Significance:** Not applicable
**Recommendation:** 6

**Clarity, Quality, Novelty And Reproducibility:**

The paper is well structured and have sufficient discussion on preliminary knowledge for readers not familiar with the field.
In terms of reproducibility I have no concerns as the authors indicate that the code will be released.

**Strength And Weaknesses:**

Strength

* The proposed method is novel and is validated on three different applications.
* The discussion of related works seems extensive.

Weaknesses

* Only qualitative analysis of the empirical result are presented.
* More ablation study is needed to support some of the claims (e.x. the complexity of numerical differentiation and automatic differentiation w.r.t. input and output dimensions).

**Summary Of The Paper:**

In this paper, the authors proposed a novel quantum transport simulator, AD-NEGF, that leverages automatic differentiation with Non-Equilibrium Green Function method. The proposed model uses implicit gradient method to accelerate backward gradient computation and is validated via different application scenarios against numerical differentiation and black-box optimization.

**Summary Of The Review:**

The paper presents a novel quantum transport simulator which is end-to-end differentiable. The proposed model is validated via experiments on three different applications and the results seem to support the claims. Despite that the direction of this paper does not entirely fit in the topics of ICLR, I think the quality of the paper meet the bar for this venue.

---

> ### Author Response · Authors · 2022-11-19
> **Response to Reviewer 4fBj**
>
> Thank you for your inspiring and helpful comments and suggestions.
>
> - **Only qualitative analysis of the empirical result is presented, and more ablation study is needed:** we have performed extra quantitative experiments to evaluate the scalability and runtime of the proposed model, compared against traditional numerical differentiation. Please refer to Appendix C in the revised manuscript.
> - **Reproducibility:** we would like to make a minor correction that, the code is already attached in the supplementary materials. We welcome cross check for the validity of the proposed method, possible directions to enhance its functionality, and more practical applications to further utilize it.
> - **The direction of this paper does not entirely fit in the topics of ICLR:** since this is a common concern raised by all reviewers, please check our post entitled "Discussion on whether this work suits ICLR 2023" for our response and for better discussion.

---

### Official Review · Reviewer_FLBL · 2022-10-28

**Confidence:** 3
**Clarity, Quality, Novelty And Reproducibility:** The paper writing is clear and easy t…
**Correctness:** 4
**Technical Novelty And Significance:** 2
**Empirical Novelty And Significance:** 3
**Recommendation:** 6

**Details Of Ethics Concerns:**

No ethics concerns.

**Strength And Weaknesses:**

Strength
1. A automatic differentiable simulator is useful for scientific study of the quantum transport.

Weakness:
1. There is no theoretical novelty of the proposed simulator. It is expected that the gradient based optimization outperforms the BO and Genetic algorithm. ICLR may not be a suitable venue for the work.
2. More discussions on the scalability and runtime is needed to understand the efficiency of the simulator.

**Summary Of The Paper:**

The paper study the problem of quantum transport simulator. It implement an automatic differentiable quantum transport simulator which can provide gradients for sensitivity analysis and solving inverse problems.

**Summary Of The Review:**

A differentiable simulator for quantum transport.

Thanks for the response. I agree that the work is under the category of AI4Science and is a good infrastructure. So I would like to increase score to 6.

---

> ### Author Response · Authors · 2022-11-19
> **Response to Reviewer FLBL**
>
> Thank you for your constructive and helpful comments and suggestions.
>
> - **No theoretical novelty:** we would like to respond to this point in two folds.
>     - Firstly, it is not technically trivial to implement a differentiable NEGF solver. We designed and implemented a customized backpropagation method through the Fast Multipole Method [1] which is used to compute the electrostatic potential from the electron density. This avoids the conventional numerical Poisson's equation solving and reduces the computational complexity of backpropagating through the electrostatic potential computation from $O(N^3)$ to $O(N^{4/3})$. Self-consistent iterations are involved multiple times in NEGF, including the self-energy computation and the electrostatic potential computation, which will be extremely memory-consuming if the entire numerical process is stored in the computational graph. To compensate for this, implicit gradient techniques [2] are utilized to avoid recording and unrolling the forward numerical iterations. Even in implementing the NEGF method itself, there are many numerical techniques involved, such as the Lopez-Sancho algorithm [3] and Pulay mixing [4] to improve the convergence of self-consistent iterations, the contour integral for the Fermi-Dirac function, etc.
>     - Secondly, the novelty in implementing differentiable physics models can only lie at the algorithmic level, focusing on improving its numerical efficiency and stability. This is because, the forward model structure is pre-determined by the underlying physics and the corresponding numerical methods, while the theoretical foundation for differentiating a physics model is the adjoint sensitivity method, which is maturely developed long ago that even for very influential works on differentiable physics models such as Neural ODE, the relevant citation dates back to the 1960s.
> - **It is expected that the gradient-based optimization outperforms the BO and Genetic algorithm:** indeed. The contribution of this work lies in building up a differentiable quantum transport solver. We choose to use simple gradient-based optimization methods by intention because, in this way, it can really show the effectiveness and power of the differentiable solver.
> - **More discussions on scalability and runtime:** we have performed extra quantitative experiments to evaluate the scalability and runtime of the proposed model, compared against traditional numerical differentiation. Please refer to Appendix C in the revised manuscript.
> - **ICLR may not be suitable:** since this is a common concern raised by all reviewers, please check our post entitled "Discussion on whether this work suits ICLR 2023" for our response and for better discussion.
>
> References
> [1] Engheta, Nader, et al. "The fast multipole method for electromagnetic scattering computation." IEEE Transactions on Antennas and Propagation 40 (1992): 634-641.
> [2] Bai, Shaojie, J. Zico Kolter, and Vladlen Koltun. "Deep equilibrium models." Advances in Neural Information Processing Systems 32 (2019).
> [3] Sancho, MP Lopez, et al. "Highly convergent schemes for the calculation of bulk and surface Green functions." Journal of Physics F: Metal Physics 15.4 (1985): 851.
> [4] Wikipedia contributors. "DIIS." Wikipedia, The Free Encyclopedia. Wikipedia, The Free Encyclopedia, 5 May. 2021. Web. 17 Nov. 2022.

---

### Official Review · Reviewer_5rbi · 2022-11-07

**Confidence:** 3
**Correctness:** 2
**Technical Novelty And Significance:** 2
**Empirical Novelty And Significance:** Not applicable
**Recommendation:** 3

**Clarity, Quality, Novelty And Reproducibility:**

Clarity: The paper is well-written in terms of use of English; however, it heavily lacks clarity given the average knowledge of the ICLR conference. It is not clear to the reviewer what is the problem at hand, what are the challenges, how this problem is related to the problems mostly highlighted in the ICLR community (and its history). The authors could have done a better job restructuring the paper so that it feels more ML/AI (mathematical computer science/ECE/statistics/applied math) rather than focusing on the material science perspective.

Quality: The quality cannot be confidently derived given the lack of expertise by the reviewer.

Novelty: Similar as quality

Reproducibility: The authors provide code, so I consider this reproducible.

**Strength And Weaknesses:**

Strength:
- (I could not identify the strengths of the paper ---other than the contributions stated--- since it is not easy to parse the paper. See below).


Weaknesses:
- The paper does not read as a great fit for the ICLR conference. While the conference is inclusive to new ideas and techniques, the paper focuses on the quantum transport simulation problem, and studies again the non-equilibrium Green function. The reviewer strongly believes that the paper would make a great fit into a more quantum-related conference. It is not clear to the reviewer how this paper would be of relevance to the audience of ICLR (which is mostly focusing on ML/AI topics). This definitely is a negative point for this paper (which might be unfair for its content).
- To the best of the reviewer's understanding, the introduction uses keywords and language that are not known to the average ICLR reviewer: E.g., a sentence that is hard to parse is "The entire numerical process of the quantum transport simulation is implemented in PyTorch, including the computation of the self-energy, the Green function, the electrostatic potential, the transport properties, as well as an optional Slater-Koster Tight-Binding (SKTB) module to generate the block tri-diagonal Tight-Binding (TB) Hamiltonian (Klymenko et al., 2021)." It is not easy for most of the reviewers in ICLR to follow this language in order to appreciate the contributions of this work. That's why the reviewer strongly believes that this paper should resubmitted to a more appropriately selected venue. See also "We propose and implement AD-NEGF, as far as we know the first end-to-end differentiable quantum transport simulator, including the NEGF method, the Poisson’s equation module for self-consistent electrostatic potential computation, and the SKTB module to generate the tight-binding Hamiltonian from the coordinates and properties of the system atoms".
- While the reviewer believes that the math is correct, he/she could not follow the math steps in equations eq. (1)-(4), what they mean, how they are derived and how these are computationally difficult to compute, etc. While introducing new optimization problems in ML/AI venues is more than welcome, that should also come in the form that is easily understandable by the corresponding audience. In other words, it could be the case that the paper could be written in a way that introduces an abstract problem and its efficient solution, and then connect it with the problems considered from the physics perspective.
- All experiments and results are better suited to a physics or materials science or systems/hardware electrical engineering venue, rather than ICLR. They are not easily appreciated for the problems considered in the current draft.

**Summary Of The Paper:**

****This is an emergency review for the paper 2820****

The paper is about Automatic Differentiation (AD) based quantum transport simulator, and the paper proposes the AD-NEGF algorithm. Key ingredients of the new algorithm efficient gradient calculation by utilizing automatic differentiation and implicit layer techniques and guarantees of correctness of the forward simulation.  The algorithm is used to calculate differential physical quantities and solving inverse problems that are intractable by traditional optimization methods.

**Summary Of The Review:**

Out of scope for ICLR. While this should be used sparely (and I'm not comfortable either using it), I find it hard to place this paper in a session in ICLR. Further, the emergency of this review definitely harms the review quality, but I feel strongly that the same outcome would be even if I had a couple more days to review this paper. The idea of automated differentiation for quantum optimal transport could be interesting in other communities, but does not feel a good fit for ICLR. This justifies this score.

---

> ### Author Response · Authors · 2022-11-19
> **Response to Reviewer 5rbi**
>
> Thank you very much for willing to respond to the call for emergency review and to provide your valuable and detailed comments.
>
> - **The paper does not read as a great fit for the ICLR conference:** since this is a common concern raised by all reviewers, please check our post entitled "Discussion on whether this work suits ICLR 2023" for our response and for better discussion.
> - **The introduction is not easy to follow for most ICLR reviewers:** we notice that this is a common concern by several reviewers, therefore provide detailed explanations in the post entitled "Introduction to the Problem". We have also updated the manuscript accordingly in order to clarify the problem settings. Please refer to the post and the revised manuscript, and provide your valuable suggestions if possible.
> - **The math is hard to follow and why these are computationally difficult:**
>     - We apologize for the confusion brought by the mathematical formulae. Because of the page limit, we can only provide a brief introduction to NEGF in the main text. Equation (1) is the standard green function expression $G=[EI-H]^{-1}$. However, since the NEGF is derived based on the assumption that the electrodes are infinitely large, computing the above green function with the full system Hamiltonian means inversion of an infinitely large matrix, making it intractable. To solve the problem, Equation (2) is proposed to replace (1). As explained with Equation (10)-(14) in Appendix A.1, if the full Hamiltonian $H$ is block-tridiagonal, we can compute the device Hamiltonian $H_D$ (which is precisely the part we care) with corrected terms called electrode self-energy $\Sigma$. Computing the self-energy relies on Equation (3), the surface green function $g_s$, which is the Green function of a single layer of the electrode that is connected to the device. Since the electrode is constructed layer by layer, again the Hamiltonian of the electrode is block-tridiagonal, which leads to the self-consistent formula of Equation (3). More details of deriving the self-consistent formula of $g_s$ can be found in [1, Chap. 7.2]. Equation (4) is the modified Poisson equation to obtain the electrostatic potential distribution, which we explain in detail in Appendix A.2. Please also refer to [2, Chap. 3.5] for more systematic discussions.
>     - The difficulties of performing such simulations and the corresponding backpropagation lie in the following aspects in major. First of all, the scale of the problem is huge. The Hamiltonian of the system is of size $N_A^2$, where $N_A$ is the number of atom orbitals. Second of all, numerical computations such as integration and PDE solving are involved, where we utilize techniques such as the contour integral and the fast multipole method to improve the numerical efficiency. Thirdly, several self-consistent iterations are involved, such as the computation for the self-energy term and the electrostatic potential, which results in some expensive computation such as matrix inversion and PDE solving to be performed many times, for which we utilize techniques such as the Lopez-Sancho algorithm [3] and Pulay mixing [4] to improve its convergence. Even more, the aspects mentioned above are just for the forward simulation. The techniques we designed and applied to backpropagate through the forward simulation are described in Section 4 of the paper.
>
> References
> [1] Pourfath, Mahdi. The non-equilibrium Green's function method for nanoscale device simulation. Vol. 3. Vienna: Springer, 2014.
> [2] Harb, Mohammed Aziz. Scattering effects in atomistic quantum transport simulations. McGill University (Canada), 2019.
> [3] Sancho, MP Lopez, et al. "Highly convergent schemes for the calculation of bulk and surface Green functions." Journal of Physics F: Metal Physics 15.4 (1985): 851.
> [4] Wikipedia contributors. "DIIS." Wikipedia, The Free Encyclopedia. Wikipedia, The Free Encyclopedia, 5 May. 2021. Web. 17 Nov. 2022.

---

### Official Review · Reviewer_uefF · 2022-11-09

**Confidence:** 2
**Correctness:** 3
**Technical Novelty And Significance:** 3
**Empirical Novelty And Significance:** 3
**Recommendation:** 5

**Clarity, Quality, Novelty And Reproducibility:**

Clarity: It might be that the paper would be clear to someone more familiar with the subject matter, but to the general ML audience or even those with limited familiarity with quantum physics that paper is difficult to read.

Quality: The comparison to prior works seems lacking. It's not well communicated what are the main methods this work should be compared against and how well it performs.

Novelty: To the extent of my limited understanding, it appears to be a novel application of auto-diff through a physical simulation to solve an intriguing problem.

Reproducibility: The code is provided, so it should probably be reproducible, but I didn't try running it.

**Strength And Weaknesses:**

Strengths:
* The first use of autodiff for solving the quantum transport problem that can alleviate many of the problems of earlier approaches.
* Introducing a new topic for the ML community.

Weaknesses:
* The problem this paper aims to solve is too obscure for the ML audience. It would not be a problem if it were introduced more clearly in order to make it more accessible. As a reader with a bit of background in quantum physics but no familiarity with the quantum transport problem, I cannot say that I have a clear understanding of what exactly the authors are trying to solve.
* The comparison to prior works is also a bit unclear. What exactly are black-box methods in this context? It would be useful to provide specific citations to what you are comparing against. Also, What is considered today to be the best performing method and how it compares? Finally, considering that one of the claimed benefits of your method is efficiency, comparing FLOPs or runtime would be crucial. (Beyond the time plot in Figure 6)
* A more minor note is the overuse of initials. It's already hard to follow a new subject, but having so many unfamiliar intials makes reading the paper even harder.

**Summary Of The Paper:**

The paper proposes to use auto-diff to compute the gradients of the Non-Equilibrium Green Function (NEGF) to solve the quantum transport problem. Numerical differentiation methods are more expensive and can result in numerical errors that could lead to wrong estimations of the physical properties of the system. Moreover, while using Neural Networks as surrogate models is a possible path to achive faster solutions, but because they are trained from data, there's no guarantee that the derived results will be accurate. The paper moves on to briefly describe the NEFG method and specific techniques they employ to compute its derivatives. Finally, the AD-NEGF method is compared against numerical differentiation, showing the correctness of AD-NEGF on one side, while demonstrating the possible pitfalls of relying on numerical differentiation. In addition to establishing the basic benefits of the proposed method, additional experiments are carried out on other settings within this domain.

**Summary Of The Review:**

My main concern is that the paper might not be a good fit for the ICLR audience unless a major revision is provided to make it more accessible. While the proposed method belongs to the intersection of ML and Physics and so it could in principle find an audience in ICLR, more space should be spent on introducing the problem to those new to it.

---

> ### Author Response · Authors · 2022-11-19
> **Response to Reviewer uefF**
>
> Thank you very much for your detailed and helpful comments and suggestions.
>
> - **The problem this paper aims to solve is too obscure for the ML audience.** we notice that this is a common concern by several reviewers, therefore provide detailed explanations in the post entitled "Introduction to the Problem". We have also updated the manuscript accordingly in order to clarify the problem settings. Please refer to the post and the revised manuscript, and provide your valuable suggestions if possible.
> - **The comparison to prior works is also a bit unclear.** Since this work is as far as we know the first trial to apply differentiable programming in quantum transport, there are no prior works to compare against directly on the performance of the differentiable simulator itself. Therefore, we compare the performance of handling the same tasks with and without a differentiable solver.
>     - For differential quantity computation, without a differentiable solver, it needs to be done by numerical differentiation. However, the computational complexity of numerical differentiation scales linearly with the input dimension, which is frequently large. Also, the precision of numerical differentiation is limited by the trade-off between the truncation error and the round-off error. The Seebeck coefficient calculation experiment in the paper validates the above claim.
>     - For solving inverse problem such as parameter identification and parameter optimization, without a differentiable solver, black-box optimization methods need to be applied. Here by "black-box", we mean optimization methods that only feed input parameters into the simulator and obtain the corresponding performance metrics, without any extra information or knowledge of the simulator. More specifically speaking, in this work, we have applied Bayesian Optimization and Genetic Algorithm. For the transmission fitting task, since it is a $10^4$ dimensional optimization, none of the black-box methods can even work because of the curse of dimensionality. For the doping optimization task, we compare the black-box methods with AD-based optimization and demonstrate the superiority of utilizing a differentiable simulator. The implementation details for Bayesian optimization and the genetic algorithm are provided in Appendix B.
> - **Comparing FLOPs or runtime**: we have performed extra quantitative experiments to evaluate the scalability and runtime of the proposed model, compared against traditional numerical differentiation. Please refer to Appendix C in the revised manuscript.
> - **Overuse of initials**: we are sorry for the inconvenience. In the following, we make a full list of all abbreviations used in the paper and provide brief and intuitive explanations.
>     - NEGF: Non-Equilibrium Green Function [1], a widely-applied method for quantum transport calculation, by constructing the device Green function with the effect of the electrodes considered in a term called self-energy, and performing a self-consistent iteration between the transport equation and Poisson's equation to obtain the correct electric potential and current.
>     - TB: Tight-Binding [2], a widely-applied approximation model in solid-state physics, which assumes that electrons are tightly bounded to the atom they belong to, and have only limited interaction with surrounding atoms.
>     - SKTB: Slater-Koster Tight-Binding [3], a specific type of TB model with semi-empirical models and parameters.
>     - DOS: Density of States [4], the number of electron states in the system at a particular energy level.
>     - AGNR: Armchair Graphene NanoRibbon, a specific type of graphene, the basic structure of which is shown in Figure 2(a).
>     - AD: Automatic Differentiation, or algorithmic differentiation, is the technique to automatically evaluate the derivative of a computer program.
>     - FD: Finite Difference, a method for numerical differentiation.
>     - FMM: Fast Multipole Method, a numerical method to speed up the calculation of N-body problems by multipole expansion.
>     - PDE: Partial Differential Equations.
>     - ASE: an atomistic simulation package including electron transport modules.
> - **ICLR may not be suitable:** since this is a common concern of all reviewers, please check our post entitled "Discussion on whether this work suits ICLR 2023" for our response and for better discussion.
>
> References
> [1] Datta, Supriyo. "The non-equilibrium Green's function (NEGF) formalism: An elementary introduction." Digest. International Electron Devices Meeting, IEEE, 2002.
> [2] Wikipedia contributors. "Tight binding." Wikipedia, The Free Encyclopedia. Wikipedia, The Free Encyclopedia, 31 Oct. 2022. Web. 17 Nov. 2022.
> [3] Slater, John C., and George F. Koster. "Simplified LCAO method for the periodic potential problem." Physical Review 94.6 (1954): 1498.
> [4] Wikipedia contributors. "Density of states." Wikipedia, The Free Encyclopedia. Wikipedia, The Free Encyclopedia, 16 Nov. 2022. Web. 17 Nov. 2022.

---

### Author Response · Authors · 2022-11-08
**Discussion on whether this work suits ICLR 2023**

While we are intensely working on addressing the detailed comments and suggestions, which require extra thinking and experimentation, we notice that there are common concerns shared by all reviewers about if this paper suits ICLR 2023. Therefore we would like to reply to this point first, share our considerations, and hopefully have further discussions with the reviewers.

We totally understand the reviewers' concern that quantum transport is not within the traditional application scope of machine learning. However, the technical contribution we have made in this work belongs to automatic differentiation and differentiable programming, instead of further developing the quantum transport domain knowledge. Different from constructing a neural surrogate model, differentiable physics models indeed involve domain knowledge more deeply, but still serve the same purposes for better learning, control, and optimization. Such efforts have been acknowledged by top-tier AI conferences before, such as ICLR [1,2], NeurIPS [3,4,5], etc.

We are all experiencing a significant and exciting trend in AI4Science, such as differential equations, physics, materials, climate, biology, etc. AI may make a huge difference in these domains, and replicate its success in CV, NLP, robotics, etc. This is probably why ICLR 2023 has made "Machine Learning for Sciences" a tier-1 category. Taking the famous and popular AlphaFold [6] for comparison, we do not see why quantum transport should be any less important or appealing to machine learning researchers than protein folding.

References
[1] Holl, Philipp, Nils Thuerey, and Vladlen Koltun. "Learning to Control PDEs with Differentiable Physics." International Conference on Learning Representations. 2019.
[2] Ingraham, John, et al. "Learning protein structure with a differentiable simulator." International Conference on Learning Representations. 2018.
[3] Schoenholz, Samuel, and Ekin Dogus Cubuk. "JAX MD: a framework for differentiable physics." Advances in Neural Information Processing Systems 33 (2020): 11428-11441.
[4] Um, Kiwon, et al. "Solver-in-the-loop: Learning from differentiable physics to interact with iterative PDE-solvers." Advances in Neural Information Processing Systems 33 (2020): 6111-6122.
[5] de Avila Belbute-Peres, Filipe, et al. "End-to-end differentiable physics for learning and control." Advances in neural information processing systems 31 (2018).
[6] Jumper, John, et al. "Highly accurate protein structure prediction with AlphaFold." Nature 596.7873 (2021): 583-589.

---

### Author Response · Authors · 2022-11-19
**Introduction to the Problem**

Since there are common concerns about improving the introduction to better fit the machine learning community, we renew the introduction to help readers better understand the scenario and the problems we are trying to solve. We have also revised the manuscript accordingly, and sincerely look forward to your advice.

The strong and lasting demand for higher computing power and lower energy consumption urges the downscale of semiconductor devices. Over the last 40 years, the microelectronics industry has successfully made the transistor feature size scale from 10$\mu\text{m}$ to nearly 20nm, of which size the quantum mechanical effect starts to dominate. In this case, simulating such devices requires a quantum transport theory-oriented formulation, while the NEGF method, as a representative, is one of the most rigorous approaches among existing quantum transport methods [1].

Although NEGF shows superiority in simulation accuracy, it is also extremely time and computation-consuming. We want to improve the efficiency of electronic device simulations and accelerate the design process with automatic differentiation techniques. An end-to-end differentiable NEGF-based quantum transport simulator can be extremely useful in the following circumstances.

Firstly, practically useful information in device simulation is often related to calculating derivatives. For instance, the thermoelectric property measured by the Seebeck coefficient is related to the derivative of transmission function $T(E)$ with respect to the carrier energy $E$; the sub-threshold swing of MOSFET is associated with the derivative of the drain current $I_D$ with respect to the applied gate voltage $V_g$, etc. While the derivatives could also be obtained by the finite difference method (FD), its computational complexity is proportional to the input dimension, and its accuracy is bounded by the trade-off between the truncation and the round-off error when choosing the step size [2]. This is where automatic differentiation could contribute, as backward-mode AD has constant complexity dependency on input parameter size, and AD can guarantee the numerical accuracy to the up limit of the machine precision.

Secondly, electronic device simulations are conducted chiefly in order to better understand the material properties and search for appropriate device configurations. This is known as Technology Computer-Aided Design (TCAD), which provides methodologies to reduce time and expenditure [3]. In the microelectronics industry, lots of effort are spent to conduct a huge amount of simulations for the dependency of transport properties, which is the output of the simulator, with respect to the input variables, such as the number of atoms to be injected into the materials [4], device sizes [5], deformation of atom positions under certain applied forces [6], etc. Such problems are often referred to as the theoretical inverse problem of learning material parameters. The R&D efficiency can be greatly improved if we can find proper device configurations without conducting endless simulations for brute-force search. Having an end-to-end differentiable simulator will directly transform such problems into gradient-based optimization problems by providing gradient information efficiently and accurately, which is extremely useful and in fact, critical, since it makes joint control of high-dimensional parameters practical.

References
[1] Jin, S. (2006). Modelling of quantum transport in nano-scale MOSFET devices. Seoul National University, Seoul.
[2] Gautschi, W. (2011). Numerical analysis. Springer Science & Business Media.
[3] Pourfath, Mahdi. The non-equilibrium Green's function method for nanoscale device simulation. Vol. 3. Vienna: Springer, 2014.
[4] Ahmad, A. H., and A. M. Awatif. "Dopping effect on optical constants of polymethylmethacrylate (PMMA)." Engineering and Technology Journal 25.4 (2007).
[5] Radsar, Tahereh, et al. "Effects of channel dimension and doping concentration of source and drain contacts on GNRFET performance." Silicon 13.10 (2021): 3337-3350.
[6] Si, Chen, Zhimei Sun, and Feng Liu. "Strain engineering of graphene: a review." Nanoscale 8.6 (2016): 3207-3217.

---

### Decision · Program_Chairs · 2023-01-20

**Decision:**

Reject

**Justification For Why Not Higher Score:**

Overall, there was a consensus that the paper: i) is not a great fit to ICLR; ii) could be explained better or at least in a way more understandable to the ML community; iii) needs better connections to applications in the ML community.

**Justification For Why Not Lower Score:**

N/A

**Metareview: Summary, Strengths And Weaknesses:**

- Summary:

The paper proposes to use auto-diff to compute the gradients of the Non-Equilibrium Green Function (NEGF) to solve the quantum transport problem. Numerical differentiation methods are more expensive and can result in numerical errors that could lead to wrong estimations of the physical properties of the system. The paper moves on to briefly describe the NEFG method and specific techniques they employ to compute its derivatives. The AD-NEGF method is compared against numerical differentiation, showing the correctness of AD-NEGF on one side, while demonstrating the possible pitfalls of relying on numerical differentiation.

- Strengths
Findings of the paper include:
1. The first use of autodiff for solving the quantum transport problem that can alleviate many of the problems of earlier approaches;

- Weaknesses
1. Not a great fit to the ML community (at least, yet)
2. The comparison to prior works is also a bit unclear.
3. The topic, the introduction to it and the whole notation + motivation are not known to the average ICLR reviewer; This connects with weakness #1, but this bullet point also touches upon the clarity/presentation of the paper.
4. There is no theoretical novelty of the proposed simulator; more discussion on the computational aspect of the approach (complexity).

- What would be missing:
While the authors did a good job trying to explain their perspective on the points above, the proposed changes are more than minor to make a decision without another round of reviews. Overall, the authors could/should consider:
1. A better suited venue for this type of work (maybe a quantum information/algorithms conference?)
2. In case the authors are interested in following the ML path, the overall suggestion is to bring this problem closer to what ML attendees know and understand: applications that are closer to ML (the argument about AI4Science is one, but still one of the reviewers complained that there is not enough comparisons + background to understand the alternative AI4Science methodologies for this problem), notation closer to ML community, etc.